# DataSynK: Causal-Symbolic EHR Synthesis for Tabular Foundation Models in Low-Resource Settings

**Eduarda Chagas** [1] [2]   **Roberta Viola** [1] [2]   **Juarez Monteiro** [1]   **Francisco Galuppo Azevedo** [1] [2]   **Saulo Saturnino** [1] [2]
**Adriano Veloso** [1] [2]

## Abstract

The chronic scarcity of labeled electronic health records (EHRs) limits the development of tabular foundation models, especially in Global South settings. While deep generative models collapse under extreme data scarcity, traditional structured generators fail to guarantee clinical plausibility. To address this, we propose DataSynK, a novel pipeline integrating causal discovery, prior medical ontology, and symbolic logic constraints to synthesize binary tabular EHRs. Empirical evaluations on real-world clinical data show that DataSynK avoids the mode collapse exhibited by deep generators in low-resource regimes and achieves the highest ontological validity among the evaluated methods, together with the strongest balanced-classification utility. While DataSynK is not the best on every marginal-fidelity metric, it is the only method that combines positive ontological validity with the highest downstream utility, suggesting that ontology-guided structural validity is a useful complement to distributional fidelity for knowledge-guided clinical data generation. Code is available at: https://github.com/eduarda-chagas/DataSynk.

## 1. Introduction

Tabular foundation models (FMs) increasingly rely on In-Context Learning (ICL), where predictions are made in a single forward pass without task-specific optimization (Qu et al., 2025). This regime depends on large volumes of synthetically generated tables for pre-training. Deep generators such as TabDDPM (Kotelnikov et al., 2023) and GReaT (Borisov et al., 2022) are state-of-the-art but suffer mode

collapse in extreme scarcity ($n < 50$) (Afonja et al., 2023). Bayesian-network generators remain sample-efficient in this regime (Kaur et al., 2021) but lack the clinical validity required for safe downstream use.

The current literature reveals a critical methodological gap: the absence of a unified framework capable of integrating (i) the sample efficiency and prior-injection stability of BNs, (ii) principled causal structure learning with identifiability guarantees, and (iii) hard symbolic constraint enforcement to ensure clinical plausibility.

To address this gap, we propose DataSynK, a framework for generating structured binary tabular EHR data to serve as a synthetic pre-training corpus for tabular FM, with particular emphasis on the extremely low-resource regime across the Global South. Unlike text-generative approaches, DataSynK operates entirely in the structured tabular domain: its output is a binary feature matrix encoding clinical entities extracted via ontology-guided NER, suitable as a pre-training corpus for tabular foundation models.

Our contributions are as follows:

1. **Causal-Symbolic Architecture for Structural Validity**: We propose DataSynK, a sample-efficient pipeline for binary tabular EHR synthesis that integrates Prior Knowledge Graphs (PKG), tiered prior-constrained causal discovery, and Answer Set Programming (ASP) logical filters.

2. **Improved Utility and Structural Validity:** In the evaluated low-resource clinical setting, DataSynK avoids the mode collapse observed in deep generators, attaining the highest balanced-classification utility ($\Delta\mathrm{F1} = +0.093$) and strictly positive ontological validity (Onto.Val), at the cost of a modest increase in marginal distance relative to the strongest fidelity baselines.

3. **Toward Priors for Tabular Foundation Models**: We position DataSynK as a candidate source of clinically structured synthetic data for tabular foundation models, and provide downstream evidence through a TSTR proxy with TabPFN. Directly pre-training or adapting

---
[1]Kunumi Institute, Brazil [2]Department of Computer Science, UFMG, Brazil. Correspondence to: Eduarda Chagas <eduarda.chagas@kunumi.com>.

*Proceedings of the $43^{rd}$ International Conference on Machine Learning*, Seoul, South Korea. PMLR 306, 2026. Copyright 2026 by the author(s).

a tabular foundation model on DataSynK-generated data is left to future work.

## 2. Related Work

**Tabular foundation models.** TabPFN (Hollmann et al., 2023) introduced the paradigm of in-context learning for tabular classification by meta-training on synthetic data derived from structural causal priors. Although subsequent models like TabICL (Qu et al., 2025) and CARTE (Kim et al., 2024) have scaled this approach to broader feature spaces, the core reliance on data quality persists. TabPFN's inductive biases align with causal structure, suggesting that ICL generalization improves when synthetic pre-training data preserve causal mechanisms rather than statistical approximations, consistent with analogous time-series findings (Xie et al., 2025).

**Knowledge-Guided EHR Generation.** To enhance predictive utility for underrepresented cohorts, previous work has successfully employed subpopulation-specific generation (Perets & Rappoport, 2023) and knowledge-guided architectures that constrain synthetic outputs using medical ontologies (Uppalapati et al., 2025). In the realm of structurally rigorous data synthesis, the SimSUM framework (Rabaey et al., 2025) was recently proposed as a benchmark for generating synthetic EHRs via expert-crafted Bayesian Networks. While this guarantees clinical fidelity, its strict reliance on exhaustive human curation inherently limits scalability. Addressing this critical bottleneck, our proposed method automates causal graph inference through the integration of prior knowledge graphs (PKG) and strict logical constraints (ASP), achieving rigorous medical validity without the prohibitive cost of manual parameterization.

## 3. Our Contributions

### 3.1. Problem Setup

Let $\mathcal{D}_k = \{\mathbf{x}_i^{(k)}\}_{i=1}^{n_k}$ denote the dataset for clinical subgroup $k \in \mathcal{K}$, where $\mathbf{x}_i^{(k)} \in \{0,1\}^d$ encodes $d$ binary clinical features extracted from pt-BR EHRs. The index $k$ identifies a disease-age stratum with the *critical low-resource regime* defined as $n_k < 50$.

**Objective**. Learn a per-subgroup generator $G_k : \mathcal{E} \to \{0,1\}^d$ whose samples $\tilde{\mathbf{x}} \sim G_k(\varepsilon)$ simultaneously satisfy:

(i) **Statistical fidelity**: Marginal and pairwise distributions approximate $P(\mathbf{x}^{(k)})$;

(ii) **Ontological validity:** Generated samples preserve the salient pairwise clinical co-occurrences encoded in the reference structure (formalized as ONTO.VAL, Definition 4.1);

(iii) **Structural consistency**: Samples respect the tier ordering and hard clinical constraints $\mathcal{R}$ of the reference structures.

### 3.2. DataSynK

Our method for generating binary tabular EHR data, as illustrated in Figure 1, consists of a four-step process described as follows:

Step 1: **Prior Knowledge Graph Construction**. Given clinical features $\{f_1, \ldots, f_d\}$, we construct $\mathcal{G}_{\text{PKG}} = (\mathcal{V}, \mathcal{E}_{\text{prior}})$ where each directed edge $(v_i, v_j) \in \mathcal{E}_{\text{prior}}$ encodes a causal hypothesis derived from SNOMED-CT® ontology[1]. An edge is included when a clinical coding guideline specifies a prerequisite or consequential relationship. A forbidden-edge constraint is imposed for mutually exclusive entities.

Step 2: **Tiered Causal Discovery**. To extract the causal skeleton $G^*$ from the binary features, we employ a topologically constrained variant of the continuous DAG optimization of Zheng et al. (2018):

$$\min_{\mathbf{W}} \ \mathcal{L}(\mathbf{W}; \mathcal{D}_k) + \lambda \|\mathbf{W}\|_1 \quad \text{subject to}$$

$$\text{tr}\big(e^{\mathbf{W} \odot \mathbf{W}}\big) - d = 0.$$

To ensure clinical directionality and computational efficiency, features are partitioned into causal tiers **(detailed in Appendix B)**. The optimization is solved pairwise across valid tier combinations, explicitly masking biologically impossible reverse-causal edges. Here, the continuous objective serves solely as a proxy to estimate structural edge weights $\mathbf{W}$ for structural recovery, rather than generative parameters. High-confidence prior edges initialize $\mathbf{W}$, and forbidden edges are masked throughout optimization. A sparsity threshold $\varepsilon$ is applied post-optimization to extract the final boolean DAG. Features are normalized prior to optimization to mitigate the known scale-sensitivity of the NOTEARS objective (Lawrence et al., 2021). We emphasize that this procedure recovers a prior-constrained dependency structure rather than identified causal mechanisms. Throughout, terms such as "causal validity" denote consistency with the reference PKG and tier ordering, not verified physiological causation.

Step 3: **Network Parameterization and Sampling**. The learned DAG $G^*$ defines the factorization $P(\mathbf{x}) = \prod_{j=1}^{d} P(x_j \mid \mathbf{x}_{\text{Pa}(j)})$. Conditional probability tables

---

[1] SNOMED CT® is a registered trademark of the International Health Terminology Standards Development Organization (IHTSDO). Used under license.

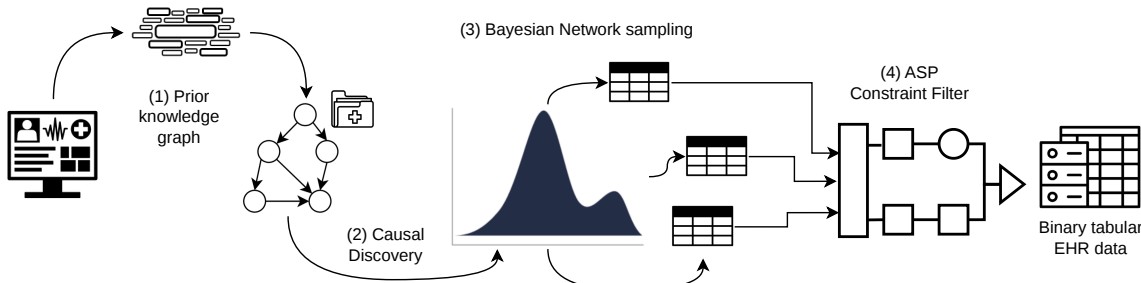

*Figure 1.* **Proposed pipeline architecture for synthetic binary tabular EHR data generation.** The process begins by integrating clinical consensus into a **(1) prior knowledge graph (PKG)**. A robust causal model, parameterized by initial EHR data and informed by the PKG, is learned via **(2) Causal Discovery**. Unconstrained synthetic patient cohorts are generated through **(3) Bayesian Network sampling**. These records are then passed through an **(4) ASP Constraint Filter**, which prunes biologically implausible cases by enforcing rigid logical rules based on clinical guidelines, ensuring the validity of the final synthetic dataset.

are estimated by maximum likelihood with Laplace smoothing $\alpha = 1/n_k$, critical for sparse parent configurations in the $n < 50$ regime. The samples are drawn by ancestral sampling over the topological ordering of $G^*$.

Step 4: **Constraint Filtering.** Raw BN samples are subjected to a hard-rejection filter implemented in Answer Set Programming. A set $\mathcal{R}$ of clinical integrity constraints, encoding parent-child code consistency and cardinality requirements, is enforced **(concrete examples of our dynamic ASP rule injection are provided in Appendix A)**: a sample $\tilde{x}$ is accepted if and only if $\tilde{x} \models r$ for all $r \in \mathcal{R}$.

## 4. Experimental Results

DataSynK is evaluated on a real-world clinical dataset derived from de-identified EHRs collected at a Brazilian public hospital within the SUS network[2]. The dataset comprises 643 records across three clinical conditions (MI, CVA, sepsis) and three age strata, yielding nine distinct subgroups with pronounced size imbalance ($n_k \in \{7, \ldots, 227\}$). All records are de-identified.

Our evaluation focuses on a head-to-head comparison against established generators using the most populous subgroups, *Adult-MI* ($n = 227$) and *Adult-CVA* ($n = 211$), which provide sufficient support for robust statistical and utility benchmarking. For these cohorts, we employ an 80/20 stratified split protocol.

---

[2]This study was conducted in compliance with Brazilian National Health Council Resolution CNS 466/12 and the General Data Protection Law (LGPD). The project was approved by the institutional Research Ethics Committee under opinion number [blinded for review] and CAAE [blinded for review], with a waiver of informed consent due to the use of a retrospective and de-identified dataset.

*Table 1.* Benchmark evaluation averaged over two primary clinical subgroups (*Adult-MI*, $n$=227; *Adult-CVA*, $n$=211). $\Delta$AUC $=$ AUC$_{\text{TSTR}}$ − AUC$_{\text{TRTR}}$ and $\Delta$F1 $=$ F1$_{\text{TSTR}}$ − F1$_{\text{TRTR}}$, where per-subgroup upper bounds are AUC$_{\text{TRTR}}$ $\in$ $\{0.785, 0.571\}$ and F1$_{\text{TRTR}}$ $\in$ $\{0.452, 0.450\}$. **Bold** indicates best result per column among non-collapsed methods. [†] Collapsed generators (TVD $> 0.30$).

| Method | TVD ↓ | \|ΔAUC\| ↓ | ΔF1 ↑ | Δcorr ↓ | Onto.Val ↑ |
|---|---|---|---|---|---|
| TRTR (real → real) | 0.000 | 0.000 | 0.000 | 0.000 | 1.000 |
| SDV | 0.015 | 0.040 | +0.040 | 0.061 | 0.052 |
| PrivBayes[†] | 0.336 | 0.094 | −0.032 | 0.078 | 0.000 |
| medGAN[†] | 0.462 | 0.160 | −0.006 | 0.082 | 0.000 |
| CTGAN | **0.009** | 0.017 | +0.013 | 0.059 | 0.062 |
| TVAE | 0.023 | **0.014** | +0.069 | **0.056** | 0.038 |
| TabDDPM[†] | 0.435 | 0.094 | −0.037 | 0.139 | 0.007 |
| **DataSynK** (ours) | 0.018 | 0.091 | **+0.093** | 0.066 | **0.089** |

**Baselines**: SDV (Patki et al., 2016), medGAN (Choi et al., 2017), CTGAN and TVAE (Xu et al., 2019), PrivBayes (Zhang et al., 2017), and TabDDPM (Kotelnikov et al., 2023). Synthetic sets generated at 1:1 ratio with training size. All metrics reported represent mean between the respective cohorts and over three generation seeds.

### 4.1. Benchmark evaluation

**Definition 4.1** (Ontology-Guided Structural Validity). For binary features $x_a, x_b$ on a dataset $D$, let $p_a = \frac{1}{|D|} \sum x_a$ and $p_{ab} = \frac{1}{|D|} \sum x_a x_b$, and define the co-occurrence lift

$$\text{lift}_D(a, b) = \frac{p_{ab}}{p_a \, p_b}, \qquad p_a, p_b > 0.$$

*Reference structure.* From the real reference set $\mathcal{D}_{\text{real}}$ we extract the set of ontology-proxy relations

$$\mathcal{R} = \{(a, b) : p_{ab} \geq \tau_c \text{ and lift}_{\mathcal{D}_{\text{real}}}(a, b) \geq \tau_\ell\},$$

i.e. feature pairs whose empirical co-occurrence in the real data is both frequent ($\tau_c = 0.01$) and strongly positively

associated ($\tau_\ell = 2.0$). These thresholds select the salient clinical co-occurrences that a faithful generator should preserve.

*Validity score.* Given a synthetic set $\widetilde{\mathcal{D}}$, a reference relation $(a, b) \in \mathcal{R}$ is *preserved* when its synthetic lift retains at least half of its real strength,

$$\text{lift}_{\widetilde{\mathcal{D}}}(a, b) \ \geq \ \tfrac{1}{2}\,\text{lift}_{\mathcal{D}_{\text{real}}}(a, b),$$

with $\text{lift}_{\widetilde{\mathcal{D}}}(a, b) = 0$ whenever $p_a = 0$ or $p_b = 0$ in $\widetilde{\mathcal{D}}$. Writing $\mathcal{P}(\widetilde{\mathcal{D}})$ for the set of preserved relations, the score is their fraction,

$$\text{ONTO.VAL}(\widetilde{\mathcal{D}}) \ = \ \frac{|\mathcal{P}(\widetilde{\mathcal{D}})|}{|\mathcal{R}|} \ \in \ [0, 1].$$

By construction $\text{ONTO.VAL} \in [0, 1]$, equal to 1 when every salient real co-occurrence is recovered, which is why the real-to-real reference (TRTR) scores $1.000$ in Table 1. A generator that destroys these associations drives the synthetic lifts below the half-strength threshold and scores near 0; the four baselines at $\text{ONTO.VAL} = 0.000$ thus recover none of the reference structure. Because $\mathcal{R}$ is induced from the real data by a lift threshold rather than verified against the external SNOMED-CT graph, $\text{ONTO.VAL}$ is conservative and partly reflects the constraints DataSynK enforces (see Limitations). The cutoff $\tau_\ell = 2.0$ and the half-strength factor are fixed a priori; a sensitivity analysis analogous to the DAG sparsity ablation (Appendix D) is left to future work.

Downstream utility was assessed via a Train-on-Synthetic-Test-on-Real (TSTR) protocol (Hyland et al., 2018) using TabPFN (Hollmann et al., 2023). Synthetic labels are assigned by $k$-NN (Cover & Hart, 1967) identically across generators, so the cross-generator ranking is driven by the synthetic feature distributions, not by method-specific labeling. Overall performance was quantified by statistical fidelity (TVD, $\Delta$corr), structural preservation (Onto.Val), and balanced predictive utility ($\Delta\text{F1} = \text{F1}_{TSTR} - \text{F1}_{TRTR}$). For mortality prediction under label imbalance, $\Delta\text{F1}$ is the clinically preferred metric as it penalizes failures on the minority class equally; $|\Delta\text{AUC}|$ is reported for completeness.

PrivBayes, TabDDPM, and medGAN deviate substantially from the real marginals. Among non-collapsed methods, CTGAN and TVAE attain lower TVD and $|\Delta\text{AUC}|$, but DataSynK is the only one simultaneously competitive on marginal fidelity, highest on $\Delta\text{F1}$ ($+0.093$), and positive on Onto.Val, a favorable fidelity-validity trade-off. This indicates that causally structured synthetic data trains more class-balanced classifiers, clinically relevant for mortality prediction under label imbalance.

### 4.2. Structural validity: What standard metrics miss?

Table 1 reveals a systematic dissociation between marginal fidelity and structural validity. For instance, while CTGAN achieves the lowest TVD, its Onto.Val remains significantly below DataSynK. DataSynK achieves positive structural preservation across the primary clinical cohorts, suggesting that our approach captures a structural dimension not reflected in standard fidelity metrics. Four baselines score Onto.Val = 0, despite three of them achieving low TVD: statistical fidelity and structural validity are dissociated. This dissociation is sharpest under the DAG sparsity threshold $\varepsilon$: loosening it marginally improves TVD while collapsing Onto.Val to zero (Appendix D), direct evidence that TVD is blind to structural and clinical degradation.

### 4.3. Privacy analysis

DataSynK does not memorize training records: its DCR and NNDR are comparable to non-collapsed baselines such as CTGAN and TVAE, with a slightly higher MIA susceptibility consistent with the fidelity-privacy trade-off (Appendix E).

## 5. Limitations.

Several limitations should be noted. First, Onto.Val is defined relative to the reference PKG, which makes it a conservative criterion and sensitive to ontology incompleteness: missing edges may penalize valid but unencoded co-occurrences, and strong Onto.Val partly reflects the constraints that DataSynK explicitly enforces. Second, our evaluation is based on a single Brazilian public-hospital cohort, limiting claims of generalization across healthcare systems. These limitations motivate multi-institutional validation, evaluation on held-out or externally curated clinical constraints, and ablations isolating the contributions of the PKG, causal discovery, and ASP filtering.

## 6. Conclusion

We introduced DataSynK, a causal-symbolic pipeline for clinically structured tabular EHR synthesis in low-resource settings. It trades a small amount of marginal fidelity for ontological validity and balanced-classification utility, a trade-off standard fidelity metrics fail to capture, positioning it as a promising source of clinically structured synthetic data for tabular foundation models.

## 7. Impact Statement

By synthesizing clinically structured EHRs without massive English-centric datasets, DataSynK offers a language-agnostic, privacy-conscious route to more inclusive clinical AI in low-resource settings.

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

## A. Implementation Details of Symbolic Logic Constraints (ASP)

In the DataSynK pipeline, raw Bayesian Network sampling can theoretically produce biologically implausible configurations due to the stochastic nature of the generation process. To enforce strict clinical guidelines (Step 4), we formalize medical knowledge using Answer Set Programming (ASP), a declarative logic programming paradigm. To optimize sample yield and computational efficiency, these ASP constraints are dynamically injected as deterministic masks during the forward sampling process. If a node evaluates to true for a constraining rule, its adjusted activation probability is forced to $P = 0.0$. Below, we provide concrete examples of the clinical rules utilized to prune invalid patient configurations, represented in standard ASP syntax.

**Causal Chain Dependencies.** Clinical events must follow a valid chronological and causal sequence. For instance, a diagnosis cannot exist without a prior underlying condition or external influence (R1), and a treatment outcome cannot be recorded without a prior diagnosis (R2). In ASP, these hard constraints are defined as integrity rules (where ':-' denotes "it cannot be the case that"):

```
% R1: A patient cannot have a diagnosis without a Tier 1 condition
:- active(Patient, Node), category(Node, diagnosis),
   not has_tier1_condition(Patient).

% R2: A patient cannot have a treatment outcome without a diagnosis
:- active(Patient, Node), category(Node, treatment_outcome),
   not has_diagnosis(Patient).
```

**Clinical and Pharmacological Conflicts.** Mutually exclusive events, such as administering a medication to a patient with a known allergy to that specific compound, are strictly forbidden (R5).

```
% R5: Prevent conflicting assignments (e.g., Medication vs. Allergy)
:- active(Patient, Medication), active(Patient, Allergy),
   represents_allergy_to(Allergy, Medication).
```

**Orphan Symptom Prevention.** To ensure structural validity, if a symptom has known clinical causes mapped in the Prior Knowledge Graph (PKG), it cannot spontaneously activate without at least one of its causal parents being active (R3).

```
% R3: An induced symptom requires at least one active parent cause
:- active(Patient, Symptom), category(Symptom, symptom),
   has_mapped_parents(Symptom),
   #count { Parent : active(Patient, Parent),
            causes(Parent, Symptom) } == 0.
```

**Cardinality and Complexity Limits.** To prevent generating clinically unrealistic "super-patients" with extreme multi-morbidities, we enforce cardinality constraints based on clinical categories. For instance, a synthetic record is constrained to a maximum of one primary diagnosis and two underlying conditions.

```
% Enforce maximum cardinality per clinical category
:- #count { Node : active(Patient, Node),
            category(Node, diagnosis) } > 1.
:- #count { Node : active(Patient, Node),
            category(Node, underlying_condition) } > 2.
```

## B. Clinical Feature Stratification and Causal Tiers

To ensure biological plausibility and prevent temporal leakage during the continuous causal discovery process the clinical features in DataSynK are strictly partitioned into a topological hierarchy. This stratification is defined by a 4-level causal tier system.

The core assumption of this topology is that clinical causality flows strictly forward through time and disease progression (from lower to higher tiers). Consequently, any structural edge projecting from a higher tier to a lower tier is mathematically masked prior to the NOTEARS optimization. The clinical semantics of each tier are defined as follows:

- **Tier 1: Baseline and Exogenous Factors (*underlying_condition*, *external_influence*).** This foundational tier encompasses pre-existing patient states and external factors that precede the current clinical episode. It includes chronic comorbidities, demographic traits, genetic predispositions, and environmental influences. By definition, these variables act as root causes and cannot be caused by acute events within the current medical encounter.

- **Tier 2: Primary Clinical States (*diagnosis*).** This tier represents the core clinical diagnoses or acute diseases identified during the patient's admission. Diagnoses are causally downstream of baseline vulnerabilities (Tier 1) and act as the primary drivers for subsequent physiological manifestations.

- **Tier 3: Manifestations (*symptom*).** This tier captures the observable clinical findings, symptoms, and physiological derangements presented by the patient. Symptoms are modeled strictly as the direct effects of the primary diagnoses (Tier 2) or underlying conditions (Tier 1).

- **Tier 4: Interventions and Results (*treatment_outcome*).** The final tier represents the culmination of the clinical pathway. It includes medical interventions, pharmacological treatments, and ultimate patient outcomes (e.g., ICU admission, survival, or mortality). These events are the downstream consequences of the patient's baseline state, diagnosis, and symptomatic presentation.

By enforcing this deterministic 4-tier topological ordering, DataSynK prevents classic statistical confounding errors, such as a symptom or death incorrectly appearing as the cause of a chronic underlying condition, while drastically reducing the computational search space for the structural optimization algorithm.

## C. Computational Efficiency: Tiered Subgraph Parallelization

A well-known limitation of continuous DAG optimization frameworks, such as NOTEARS, is their cubic time complexity $\mathcal{O}(d^3)$ with respect to the number of variables $d$. For comprehensive EHR datasets containing dozens or hundreds of clinical features, running a monolithic optimization becomes computationally prohibitive.

To overcome this bottleneck, DataSynK leverages the clinical tier structure (described in Section 3.2) to decompose the global DAG discovery into a set of highly efficient, parallelizable sub-problems. Because clinical causality is strictly unidirectional across tier, the full $d \times d$ adjacency matrix does not need to be optimized simultaneously.

Our parallelization strategy is implemented as follows:

1. **Pairwise Subgraph Decomposition:** The global feature set is partitioned into independent subgraphs representing valid causal directions between adjacent and non-adjacent tiers (e.g., Tier 1 → Tier 2, Tier 1 → Tier 3, Tier 2 → Tier 3, etc.).

2. **Dimensionality Reduction:** For each valid pair, a local NOTEARS instance is formulated containing only the nodes belonging to those two specific tiers. This reduces the effective dimensionality of each optimization routine from $d$ to $d_{sub}$, where $d_{sub} \ll d$. Consequently, the local complexity drops to $\mathcal{O}(d_{sub}^3)$.

3. **Asynchronous Parallel Execution:** Because the structural constraints strictly isolate the causal flow between these tier pairs, the subgraphs are entirely independent. We execute these local optimizations simultaneously using a multi-worker process pool, fully utilizing modern multi-core architectures.

4. **Global Recombination:** After all parallel workers converge, the local continuous weights are aggregated to reconstruct the global adjacency matrix $\mathbf{W}$. In cases where an edge is evaluated in multiple overlapping sub-contexts, the maximum learned causal strength is preserved. Finally, the global structural priors and the global sparsity threshold $\varepsilon$ are applied.

This tiered subgraph parallelization transforms a monolithic $\mathcal{O}(d^3)$ bottleneck into a scalable set of parallel $\mathcal{O}(d_{sub}^3)$ operations. This architectural choice not only accelerates the Causal Discovery step by orders of magnitude but also mathematically prevents the algorithm from wasting computational cycles searching for biologically impossible reverse-causal edges.

## D. Ablation Study: DAG Sparsity Threshold

To isolate the effect of the continuous DAG optimization on the structural validity of the generated data, we conducted an ablation study on the sparsity threshold parameter ($\varepsilon$) applied to the learned adjacency matrix $\mathbf{W}$. Table 2 reports the generation metrics on the primary Adult-MI cohort for $\varepsilon \in \{0.05, 0.10, 0.20\}$.

*Table 2.* Ablation study of the DAG sparsity threshold ($\varepsilon$). Bold indicates the optimal configuration chosen for DataSynK.

| $\varepsilon$ | TVD $\downarrow$ | $|\Delta\text{AUC}|\downarrow$ | $\Delta\text{corr}\downarrow$ | Onto.Val $\uparrow$ |
|---|---|---|---|---|
| 0.05 | 0.0180 | 0.1672 | 0.0661 | 0.0000 |
| **0.10** | 0.0175 | **0.0932** | 0.0640 | **0.0769** |
| 0.20 | **0.0172** | 0.1689 | **0.0625** | 0.0000 |

The results perfectly illustrate the "blind spot" of standard statistical metrics. For instance, increasing the threshold to $\varepsilon = 0.20$ marginally improves the statistical distance metrics (TVD drops to $0.0172$ and $\Delta$corr to $0.0625$). However, this strict threshold aggressively prunes critical causal dependencies, causing the Ontological Validity (Onto.Val) to collapse entirely to $0.0000$ and degrading downstream predictive utility ($|\Delta\text{AUC}|$). Conversely, a loose threshold ($\varepsilon = 0.05$) introduces spurious structural noise, similarly destroying ontological validity. Setting $\varepsilon = 0.10$ provides the optimal balance, preserving the true clinical co-occurrence patterns (highest Onto.Val) and maximizing predictive utility without sacrificing marginal fidelity.

## E. Comprehensive Privacy Analysis

This section provides an extended analysis of the privacy metrics summarized in Section 4.3. We evaluate the generators across three dimensions of privacy risk: Distance to Closest Record (DCR), Nearest-Neighbour Distance Ratio (NNDR), and Membership Inference Attacks (MIA-AUC).

**Distance Metrics (DCR and NNDR).**   DCR measures the minimum distance between any synthetic record and its closest real counterpart. As shown in Table 3, among non-collapsed methods, DCR values are highly comparable across DataSynK ($0.014$), CTGAN ($0.016$), TVAE ($0.016$), and SDV ($0.017$). This confirms that DataSynK generates novel patient representations rather than outputting near-exact copies of the training records. NNDR complements this by assessing the dispersion of synthetic samples; DataSynK's NNDR ($0.567$) aligns tightly with CTGAN ($0.570$), indicating a healthy diversity of generated cases.

**Membership Inference (MIA-AUC).**   This metric evaluates how easily an attacker can distinguish synthetic records from real ones, where an AUC $\approx 0.5$ indicates ideal indistinguishability. DataSynK achieves an MIA-AUC of $0.619$, marginally higher than CTGAN ($0.553$) and SDV ($0.583$). This slight increase is a direct consequence of the fidelity-privacy trade-off because DataSynK strictly adheres to logical medical constraints (Onto.Val) and preserves predictive causal structures ($\Delta$F1), its generated distributions are highly realistic, making it slightly more susceptible to membership inference than purely statistical (and clinically invalid) approximations.

**The Artifact of Collapsed Models.**   Collapsed generators (PrivBayes, TabDDPM, medGAN) exhibit artificially high DCR values (e.g., $0.441$ for medGAN) and NNDR values ($\approx 0.98$). However, their MIA-AUC scores approach $1.000$, confirming they are trivially distinguishable from real data. This demonstrates that their high distance from training records is merely an artifact of generating completely invalid, low-fidelity records (as evidenced by their high TVD), rather than a genuine privacy-preserving property.

*Table 3.* Privacy metrics for the primary evaluation cohorts. DCR ↑ and NNDR ↑: higher = more distant from training records. MIA-AUC: closer to 0.5 = harder to distinguish from real data. [†] Collapsed generators (TVD > 0.30).

| Method | DCR ↑ | NNDR ↑ | MIA-AUC ≈ 0.5 |
|---|---|---|---|
| SDV | **0.017** | **0.637** | 0.583 |
| CTGAN | 0.016 | 0.570 | **0.553** |
| TVAE | 0.016 | 0.518 | 0.587 |
| **DataSynK** (ours) | 0.014 | 0.567 | 0.619 |
| PrivBayes[†] | 0.317 | 0.971 | 0.999 |
| TabDDPM[†] | 0.368 | 0.967 | 1.000 |
| medGAN[†] | 0.441 | 0.981 | 1.000 |

