# OpenReview forum: "DataSynK: Causal-Symbolic EHR Synthesis for Tabular Foundation Models in Low-Resource Settings"
_ICML.cc/2026/Workshop/FMSD — FMSD @ ICML 2026 Poster_

### Official Review · Reviewer_S5vA · 2026-05-14
**Interesting Causal-Symbolic Pipeline, but Claims Outpace Evidence**

**Rating:** 5
**Confidence:** 4

**Review:**

## Summary

This paper proposes DataSynK, a pipeline for generating binary tabular EHR data in low-resource settings. The method combines a prior medical knowledge graph, constrained causal discovery, Bayesian Network sampling, and ASP-based symbolic filtering to enforce clinical constraints. The authors evaluate DataSynK on de-identified Brazilian EHR data and compare it against several synthetic tabular data generators. They report that DataSynK achieves the highest downstream ∆F1 and positive ontological validity among the evaluated methods.

## Strengths

**The paper addresses an important problem.**
Low-resource clinical data generation is a meaningful and high-impact setting, especially for underrepresented healthcare systems.

**The method is conceptually interesting.**
Combining causal priors, Bayesian Networks, and symbolic clinical constraints is a reasonable approach for small-data EHR synthesis. The pipeline is also more interpretable than black-box deep generators.

**The focus on clinical validity is valuable.**
The paper correctly argues that marginal fidelity metrics can miss clinically implausible combinations. Evaluating ontology/logical validity is a useful direction for clinical synthetic data.

**The paper includes relevant baselines.**
The comparison includes SDV, PrivBayes, medGAN, CTGAN, TVAE, and TabDDPM, giving useful context against common synthetic tabular data generators.

## Areas for Improvement

**The main evaluation does not fully match the low-resource motivation.**
The paper emphasizes extreme low-resource settings, including regimes with (n < 50), but the main benchmark uses the two largest cohorts, Adult-MI with (n=227) and Adult-CVA with (n=211). This weakens the claim that DataSynK overcomes failures in the extreme low-resource regime.

**The tabular foundation-model pretraining claim is not directly tested.**
The paper motivates DataSynK as a source of synthetic pretraining data for tabular foundation models. While it uses TabPFN in a TSTR evaluation, it does not actually pretrain or adapt a tabular foundation model using DataSynK-generated data. Thus, the experiments support downstream utility more directly than the pretraining claim.

**The ontological-validity result is useful but somewhat expected.**
Because DataSynK explicitly enforces ontology/ASP constraints, strong Onto.Val performance is an important sanity check that the constraint mechanism works. However, to show broader superiority over baselines, the paper should also evaluate clinical validity using external or held-out constraints, clinician review, or downstream real-label utility.

**The empirical superiority claim should be softened.**
DataSynK achieves the highest ∆F1 and positive Onto.Val in Table 1, but it is not best on TVD, (|\Delta AUC|), or ∆corr. Some deep baselines such as CTGAN and TVAE do not collapse and outperform DataSynK on several standard metrics. The result should be framed as a trade-off rather than broad empirical superiority.

**The downstream utility setup needs more justification.**
The TSTR evaluation uses TabPFN, but the synthetic labels are inferred using k-NN. This labeling step could strongly affect downstream F1 and AUC. The paper should justify this design and test whether the conclusions hold under alternative labeling or real-label settings.

**The causal claims are too strong.**
The method uses prior constraints and NOTEARS-style structure learning, but this does not establish that the learned graph captures true clinical causality from small observational EHR samples. Terms such as “causal validity” and “preserved causal mechanisms” should be used more cautiously.

## Detailed Comments

The main additions I would like to see are evaluations on the truly low-resource cohorts, clearer definition and external validation of Onto.Val, and ablations that isolate the contribution of the prior knowledge graph, causal discovery, and ASP filtering. To demonstrate superiority beyond rule compliance, the authors should evaluate on held-out clinical constraints, external clinician plausibility ratings, or more downstream real-label prediction tasks. I would also encourage the authors to narrow the framing: the current experiments show that DataSynK can improve one downstream F1 metric while enforcing clinical constraints, but they do not establish it as a general pretraining corpus solution for tabular foundation models in low-resource healthcare.

## Justification of Score

The topic is relevant and the causal-symbolic direction is interesting. However, the evidence does not fully support the strength of the claims. The paper emphasizes extreme low-resource EHR synthesis and tabular foundation-model pretraining, but the main evaluation uses the two largest cohorts and does not actually pretrain or adapt a tabular foundation model. The strongest validity result is meaningful, but also closely aligned with constraints that DataSynK explicitly enforces. Overall, this is an interesting idea, but the empirical support is too narrow and the framing is too broad.

---

### Official Review · Reviewer_nFMm · 2026-05-21
**EHR Synthesis in Low-Resource Settings**

**Rating:** 6
**Confidence:** 2

**Review:**

Summary: DataSynK generates synthetic binary EHR tables by combining SNOMED-CT knowledge graphs, tiered causal discovery using a constrained version of NOTEARS, Bayesian network sampling, and ASP logic filtering to remove biologically implausible records. Tested on 643 records from a Brazilian hospital across MI, CVA, and sepsis cohorts, it is compared against six generators on the two largest subgroups (n around 200 each). It is the only method achieving positive ontological validity while also delivering the highest balanced classification utility.

Strengths: The pipeline is conceptually well motivated. Ontology priors constrain graph structure, causal tiers prevent nonsensical edges like symptoms causing baseline conditions, and ASP catches biologically impossible samples after the fact. The sparsity threshold ablation in Appendix D is a good illustration of why standard metrics like TVD are insufficient.